# MARBLE: Music Audio Representation Benchmark for Universal Evaluation

**Ruibin Yuan**[*,1,2*] **Yinghao Ma**[*,3*] **Yizhi Li**[*,4,5*] **Ge Zhang**[*,1,6*] **Xingran Chen**[7] **Hanzhi Yin**[2]
**Le Zhuo**[8] **Yiqi Liu**[4,5] **Jiawen Huang**[3] **Zeyue Tian**[9] **Binyue Deng**[10] **Ningzhi Wang**[3]
**Chenghua Lin**[4,5†] **Emmanouil Benetos**[3] **Anton Ragni**[5] **Norbert Gyenge**[5] **Roger Dannenberg**[2]
**Wenhu Chen**[6] **Gus Xia**[11,12] **Wei Xue**[9] **Si Liu**[8] **Shi Wang**[13] **Ruibo Liu**[14] **Yike Guo**[9] **Jie Fu**[1†]

[*]Multimodal Art Projection Research Community   [1]Beijing Academy of Artificial Intelligence
[2]Carnegie Mellon University   [3]Queen Mary University of London   [4]University of Manchester
[5]University of Sheffield   [6]University of Waterloo   [7]University of Michigan Ann Arbor
[8]Beihang University   [9]Hong Kong University of Science and Technology   [10]Peking University
[11]New York University   [12]MBZUAI   [13]ICT, Chinese Academy of Sciences   [14]Dartmouth College
ruibiny@alumni.cmu.edu   yinghao.ma@qmul.ac.uk   yizhi.li@hotmail.com
gezhang@umich.edu   chenghua.lin@manchester.ac.uk   fujie@baai.ac.cn

## Abstract

In the era of extensive intersection between art and Artificial Intelligence (AI), such as image generation and fiction co-creation, AI for music remains relatively nascent, particularly in music understanding. This is evident in the limited work on deep music representations, the scarcity of large-scale datasets, and the absence of a universal and community-driven benchmark. To address this issue, we introduce the **M**usic **A**udio **R**epresentation **B**enchmark for universa**L** **E**valuation, termed MARBLE. It aims to provide a benchmark for various Music Information Retrieval (MIR) tasks by defining a comprehensive taxonomy with four hierarchy levels, including acoustic, performance, score, and high-level description. We establish a unified protocol based on 18 tasks on 12 public-available datasets, providing a fair and standard assessment of representations of all open-sourced pre-trained models developed on music recordings as baselines. MARBLE offers an easy-to-use, extendable, and reproducible suite for the community, with clear statements on dataset copyright. Results suggest that recently proposed large-scale pre-trained musical language models perform the best in most tasks, with room for further improvement. The leaderboard and toolkit repository are published[34] to promote future music AI research.

## 1 Introduction

Despite Artificial Intelligence (AI) rapid advancement in the field of art, it has not yet made significant progress in music, particularly in music understanding. To address this, researchers are studying the interdisciplinary field of Music Information Retrieval (MIR) to develop a general music understanding model. MIR focuses on automatically extracting information from raw music audio [40], which enables a variety of tasks such as music classification, emotion recognition, pitch estimation, and the analysis of musical features such as rhythm, melody, and harmony. Due to issues such as copyright and annotation costs, labelled music datasets are usually small, which limits the performances of

---

[*]The authors contributed equally to this work.
[†]Corresponding authors.
[3]https://marble-bm.shef.ac.uk
[4]https://github.com/a43992899/MARBLE-Benchmark

37th Conference on Neural Information Processing Systems (NeurIPS 2023) Track on Datasets and Benchmarks.

supervised models. Given that self-supervised learning (SSL) is useful for various tasks (*e.g.*, NLP [23, 18, 51] and CV [41]) with limited annotated datasets, there have been works on SSL-based audio representation learning [26, 33, 32, 2, 20, 49, 58] and music pre-trained models [43, 35, 69, 62, 31, 11, 29, 53, 66, 37]. The existing benchmarks, GLUE [57], SuperGLUE [56], and ERASER [10] in NLP, along with VTAB [68] and VISSL [16] in CV, all play an active role in promoting the development of SSL-related research topics in the corresponding domains. However, there are only scattered and fragmented evaluations of the existing music models rather than comprehensive benchmarks, making it difficult to objectively compare and draw insights across techniques.

In the current context, the SSL music systems are evaluated with downstream task datasets, including genre classification [26, 35, 69, 62, 31, 8, 29, 66, 37, 34], emotion classification [35, 31, 8, 29, 37, 34], instrument classification [35, 62, 49, 37, 34], music tagging [69, 31, 8, 53, 43, 29, 66, 37, 34], key detection [31, 8, 29, 37, 34], music detection [49], beat tracking [34] and cover song detection [66]. Existing works usually conduct evaluations with different experimental setups, and few of them explore sequential tasks such as beat tracking and source separation. Although in similar domains, SUPERB [65] and HEAR [54] are proposed to facilitate unified analysis of the learned representations of speech and sound events, the distribution of musical audio is significantly different. Thus, there is an urgent need to construct comparable, extensive, and easy-to-use benchmarks to enhance the development of music SSL.

In this paper, we propose a Music Audio Representation Benchmark for universaL Evaluation (MARBLE) to address this problem. MARBLE aims to examine the full spectrum of model capabilities, and thus proposes a taxonomy adapted from Dai et al. [9] to categorise MIR tasks, including acoustic, performance, score, and high-level description. The four-level hierarchy aligned to musician consensus serves as a guideline to further organise the datasets and helps to identify a diversified set of downstream tasks. We select popular tasks in the (now defunct) Music Information Retrieval Evaluation eXchange (MIREX) Challenge[5], and use the corresponding public datasets with limited annotations. As demonstrated in Tab. 1, the current version of MARBLE contains 18 downstream tasks, spread over 13 task categories on 12 publicly or commercially available datasets. Except for the common classification tasks, we also integrate the missing piece of the puzzle – sequence labelling tasks that require frame-wise prediction, including source separation and beat tracking. The datasets used in MARBLE are ensured easy-to-access: all datasets are available for download directly from the official repository or an external website for downloading a specific version.

In addition, we design a unified protocol and build tool-kits to evaluate the generalisation ability of the models. In MARBLE protocol, the models are regarded as backbones to provide universal representations for all tasks, and task-specific prediction heads are concatenated to further trained under *unconstrained*, *semi-constrained*, and *constrained* settings, which is defined by whether the training hyperparameters are restricted and whether the backbone model is frozen (cf. § 3.2). The evaluation suite provides codes for dataset preprocessing and examples of evaluating existing popular SSL models in the benchmark. We select 7 representative music SSL models as our baselines (cf. § 3.1) and release the evaluation results at our publicly available leaderboards[6] as a reference.

Our key contributions are listed as follows: (1) providing a diversified music understanding benchmark with well-defined taxonomy of the MIR tasks; (2) incorporating and organising a wide range of datasets to facilitate comprehensive music model evaluation; (3) designing a unified assessment protocol and building corresponding evaluation suites for processing, training, and benchmarking.

## 2   Benchmark Tasks

As demonstrated in Tab. 1, we collect datasets in MARBLE to provide the community with a standard, general-purpose, easy-to-use benchmark for various tasks covering all aspects of music. Generally, music processing involves discriminative and generative tasks. The discriminative tasks either classify or regress musical recordings as a whole or use a seq2seq model to make frame-by-frame decisions on entire sequences. The generative tasks include audio synthesis and music composition. For the initial release of MARBLE, we focus on discriminative tasks, and generative tasks are currently outside our scope. The task collection is guided by the principles of (1) receiving a high level of interest

---

[5]https://www.music-ir.org/mirex/wiki/MIREX_HOME

[6]Considering potential legal constraints, MARBLE allows to submit results on the tasks partially (*e.g.*, tasks on commercially available datasets) for the future participants.

in the MIR community, (2) having publicly available datasets allowing everyone to participate, and (3) limited labelled data to effectively measure the universality of the model. Four aspects of music are studied through 18 proposed tasks: **High-level description tasks** including key detection, music tagging, classification gender, and emotion recognition; **Score-level tasks** including estimating the pitch of a musical note, tracking beats, extracting melody, estimating the chords, and transcribing the lyrics; **Performance-level tasks** including detecting musical ornaments or techniques; and **Acoustic-level tasks** including singer identification, instrument classification, and source separation that focus more on raw audio information.

Table 1: The Dataset, Commercial License, and Prediction Head of Each Task Used for the MARBLE Benchmark. SDR refers Source-to-distortion Ratio.

| Taxonomy | Task Type | Task & Annotation | Prediction Type | Evaluation Metrics | Commercially Available |
|---|---|---|---|---|---|
| **High-level Description** | Key Detection | Giantsteps key [25] | Multi-class | Weight Score [45] | Yes |
| | Music Tagging | MagnaTagATune [28] | Multi-label | ROC-AUC & PR-AUC/AP | - |
| | | MTG Top50 [7] | Multi-label | ROC-AUC & PR-AUC/AP | - |
| | Genre Classification | GTZAN [55] | Multi-class | Accuracy | - |
| | | MTG Genre [7] | Multi-label | ROC-AUC & PR-AUC/AP | - |
| | Emotion Detection | Emomusic [52] | Regression | $R2^{\text{Valence}}$ & $R2^{\text{Arousal}}$ | - |
| | | MTG MoodTheme [7] | Multi-label | ROC-AUC & PR-AUC/AP | - |
| **Score-level** | Pitch Classification | Nsynth [14] | Multi-class | Accuracy | Yes |
| | Beat Tracking | GTZAN Rhythm [55] | Seq2Seq, Binary-class | F-measure (Threshold 20ms) | - |
| | Melody Extraction | MelodyDB [4] | Seq2Seq, Multi-class | Accuracy | - |
| | Chord Estimation | GuitarSet [63] | Seq2Seq, Multi-class | 8 different `mir_eval` score | MIT |
| | Lyrics Transcription | MulJam2.0 | Seq2Seq, Multi-class | CER, WER | - |
| | | Jamendo [12] | Seq2Seq, Multi-class | CER, WER | MIT |
| **Performance-level** | Vocal Technique Detection | VocalSet [61] | Multi-class | Accuracy | Yes |
| **Acoustic-level** | Singer Identification | VocalSet [61] | Multi-class | Accuracy | Yes |
| | Instrument Classification | Nsynth [14] | Multi-class | Accuracy | Yes |
| | | MTG Instrument [7] | Multi-label | ROC-AUC & PR-AUC/AP | - |
| | Source Separation | MUSDB18 [46] | Seq2Seq, Regression | SDR | - |

## 2.1 High-level Description Tasks

**Key detection** involves predicting the scale and key pitch levels of a song. MARBLE solves this task using the Giantsteps [25] and a subset of the Giantsteps-MTG-keys dataset [27]. Giantsteps dataset contains 604 songs and is taken as our dedicated test set. Additionally, we leverage a subset of the Giantsteps-MTG-keys dataset, which contains 1077 music pieces with single-key annotations, for training and validation. Since no standardised split is available for Giantsteps-MTG, we adopt the dataset split strategy employed in [8]. Both datasets contain 2 minutes of electronic dance music covering all 12 pitch classes in major and minor, resulting in a 24-class classification task. For performance evaluation, we employ accuracy with an error tolerance metric, a weighted score metric. This metric grants partial credit for reasonable errors, such as predicting relative secondary keys when the primary key is the ground truth [45].

**Music Tagging** refers to assigning a predefined set of tags to a given song. These tags encompass various aspects such as genre, instrumentation, mood, and tempo (*e.g.*, fast), making music tagging somewhat overlap with genre classification, emotion recognition, and instrument classification. To conduct our study, we utilise two extensive datasets: MagnaTagATune (MTT) [28] and MTG-Jamendo (MTG) [7]. The MTT dataset comprises 30-second audio clips with manual annotations for tags. It consists of 25.9k clips, amounting to a total duration of 170 hours. For MARBLE, we use the Top50 tags, and adopt a conventional (12:1:3) training, validation, and test split, aligning with all baseline approaches' practices. Besides, the MTG dataset contains 55k clips, corresponding to nearly 2k hours of music. As the audio clips in this dataset may exceed 30 seconds in length, we compute multiple embeddings using a sliding window of 30 seconds and then average them to obtain an overall embedding representation. While both datasets encompass a large number of tags, we follow the customary to limit the vocabulary to the 50 most common tags in each dataset. The evaluation metrics employed for this task are the macro-average of all tag ROC-AUCs (receiver operating characteristic

- area under the curve) and the average precision (AP) / PR-AUC (precision-recall - area under the curve). These metrics provide comprehensive insights into the model's performance across all tags.

**Genre classification** aims to assign each song the most suitable genre label. This study uses two distinct datasets: GTZAN [55] and MTG-Genre. GTZAN consists of 30-second audio clips from 10 genres, making it suitable for a multi-class classification task. To assess the performance of this dataset, we report the accuracy metric. To ensure consistent evaluation, we utilise the "fail-filtered" split as described in [24] for GTZAN. The filtered dataset comprises 930 audio tracks corresponding to approximately 8 hours of music. Besides, MTG-Genre, derived from MTG-Jamendo, contains 55k tracks but focuses solely on 95 genre tags, resulting in a multi-label classification problem. We employ the ROC and AP metrics to evaluate the performance of MTG-Genre.

**Emotion Recognition** in music aims to determine the emotional content of music pieces. In our study, we utilise two distinct datasets to evaluate the performance of emotion recognition: Emomusic [52] and MTG-MoodTheme [7]. Emomusic contains 744 pieces of 45-second music clips and is annotated with valence and arousal scores. The valence represents the positivity of emotional responses, while arousal indicates emotional intensity. The official evaluation metrics for this dataset is the determination coefficient ($r^2$) between the model's regression results and human annotations of arousal and valence [52]. During inference, we split the 45-second clips into 5-second sliding windows and computed the average prediction probability as the final prediction. Since no standard dataset split is available for Emomusic, we adopt the same partitioning as [8]. It is important to note that direct comparison of the SoTA model's results with the benchmark may be challenging due to the different dataset splits. Additionally, we utilise MTG-MoodTheme, a subset of MTG-Jamendo consisting of 18.5k audio tracks annotated with 59 human emotion labels. This is a multi-label task with ROC and AP as evaluation metrics.

## 2.2 Score-level Tasks

**Pitch Classification in Music (Monophonic)** involves determining the appropriate pitch category for a given audio sample, ranging from MIDI note numbers 0 to 127 on a semitone scale. In this study, we perform pitch classification using the Nsynth dataset [14] within the music information retrieval benchmark. It comprises 340 hours of music, with each excerpt lasting 4 seconds. Since the audio recordings in this dataset are monophonic, the pitch classification task is formulated as a 128-class classification problem, covering all possible MIDI pitch categories (fundamental frequencies from 8Hz to 12.5kHz). The evaluation metric used for this task is the accuracy achieved across all audio clips.

**Beat Tracking** determines the presence of a beat and a downbeat in each frame of a given music piece. In this benchmark, we only focus on beat tracking, making it a binary-classification task[7]. An offline approach is employed for beat tracking, allowing the model to utilise frame-level information during inference. The model generates frame-by-frame predictions at a specific frequency, which are then post-processed using a dynamic Bayesian network (DBN) [6] implemented with `madmom` [5] to obtain the final result. The GTZAN Rhythm dataset [36] is used in this study. The dataset provides frame-level annotations for each music clip in GTZAN. To enhance model performance and ensure a fair comparison with the SOTA model, adjacent frames of each beat label are also labelled as beats using a label smoothing technique commonly employed in beat tracking. The model is evaluated using the `f_measure` metric implemented in `mir_eval` [45]. A prediction is considered correct if the difference between the predicted event and the ground truth does not exceed 20ms. It is important to note that while some models may have been trained on other datasets, the GTZAN-train subset is used as the training set, and GTZAN-test is used as the test set for all MARBLE submissions.

**Chord Estimation** is to recognise the temporal music chord of a given piece of music. We implemented this task as a 421-class classification including 35 types of chords on 12 different root notes, and none. More information on the chord vocabulary can be found in Appendix D. The probing model consists of an MLP with a hidden size of 512 and concludes with a fully connected output layer. There is no post-processing involved from frame-level prediction to event-level. The predictions are aligned with the token rate of the pre-trained model. Performances are measured as by 6 measures in SOTA model [22] including `root`, `majmin`, `mirex`, `thirds`, `triads`, and `sevenths`. We have

---

[7]Due to the limitation of time and the size of the dataset, tracking the time signature (*e.g.*, 4/4 metre) and downbeat is deferred to future versions with other datasets.

added two additional evaluation metrics inspired by MIREX: `majmin_inv` and `sevenths_inv`, to assess the performance of chord recognition at the level of inversions. The metrics are implemented by `mir_eval`[45]. We use the GuitarSet [63] dataset for this task. The dataset comprises 360 excerpts, each around 30 seconds, recorded by 6 players performing 30 lead sheets in two versions (comping and soloing) across 5 styles, 3 progressions, and 2 different tempos. Four audio versions are provided, and we selected the "mix" (a monophonic mixture of the original 6-channel file) for our audio collection. The dataset offers two types of chord annotations for selection, and we chose "performed chord" as our primary annotation and used "instructed chord" to substitute specific colour chords. We divided the audio into 5-second segments and allocated 5 singers into the validation and training sets while designating one singer for the test set. Out of the 5 singers, we allocated 30% to the validation set and 70% to the training set. Segments from the same song are assigned to the same set, for instance, all 5-second segments from a particular song are grouped into the training set.

**Melody Extraction** is to recognise the pitch of melody for a given music, typically pop songs. Adhering to the methodology in [59], we divided the frequency spectrum between 0 and 8000 Hz into 360 bins, treating our task as a classification problem. The probing model consists of a single-layer bidirectional LSTM with a hidden size of 512, followed by a linear layer. The predictions are aligned with the token rate of the pre-trained model, which is then resampled to match the label rate using the nearest interpolation. Performances are measured as the Overall Accuracy metric from the `mir_eval` [45] library. We use the MedleyDB [4] dataset for this task. It has 108 full tracks, collectively lasting 7.3 hours. All tracks come with three types of melody labels given at intervals of roughly 5.8 ms. For our study, we focused on the second annotation, which indicates the fundamental frequency of mixed stems. For data splitting, we followed the partitioning strategy of [59] giving 67, 15 and 26 tracks for training, validation and testing sets respectively which was achieved after omitting a redundant track from the test set in the popular split.

**Lyrics Transcription** aims to identify the linguistic content in audio recordings of singing. In MARBLE, we focus on the evaluation of multilingual lyrics transcription, an aspect that has been under-explored in the field of lyrics transcription. We perform the task using the MulJam dataset [70], which comprises 6031 songs in 6 languages: English, French, Spanish, Italian, German, and Russian. This dataset offers a rich repository of around 153k lines with lyrics annotations. The training, validation and testing sets contain 147k, 3k and 2k lines, respectively. We set a standard train/valid/test splitting and re-labelled the MulJam dataset as MulJam2.0 with more human annotation. More information can be found in Appendix E. We also use Jamendo [12] as a test set which includes English, French, German and Spanish pop songs. There are 20 songs for each language and the dataset comes with line-level human annotation for lyrics. In line with recent literature [15, 42], the backend adopts a hybrid CTC/Attention architecture design [60]. Given the task's complexity and the necessity to capture long-term dependencies, we use a transformer with 3 encoder layers and 3 decoder layers. The output from the encoder is further processed by a fully connected layer to map it to the target dimension for the CTC loss computation [17]. We also calculate a sequence-to-sequence (S2S) loss between the output from the decoder and the true lyrics text. The final loss is a balanced combination of the CTC loss and the S2S loss. For validation and testing, we employ beam search on the transformer decoder to iteratively select the best predictions. Additionally, a transformer language model is trained from the same data split to incorporate language knowledge at test time. Performance evaluation is conducted using Character Error Rate (CER) and Word Error Rate (WER). Different from all the metrics in other tasks, WER and CER values are the less the better.

## 2.3 Performance-level Tasks

**Vocal Technique Detection** task involves identifying different singing techniques within an audio clip. For this task, the MARBLE benchmark utilises the VocalSet dataset [61], the sole publicly available dataset specifically designed for studying singing techniques. This dataset comprises recordings of 20 professional singers (9 female and 11 male) performing 17 distinct singing techniques in various contexts, amounting to a total duration of 10.1 hours. Given that the audio clips are segmented into 3-second intervals, the task focuses on determining the type of technique (*e.g.*Vibrato, Straight) rather than the precise start and end times. To evaluate the performance of models, we employ Accuracy as the evaluation metric. We use a subset of 10 different singing techniques used in Yamamoto et al. [64], which contains 15 singers in the training and validation set, and 5 for the test set. Since there is no predetermined division between the training and validation sets, we assign 9 singers to the training

set and 6 singers to the validation set. It is important to note that all 3-second segments originate from the same audio recording file within the same part of the split, such as being exclusively part of the training set. Detailed data partitioning can be found in our provided code.

## 2.4 Acoustic-level Tasks

**Instrument Classification** refers to the multi-label or multi-class identification of instruments present in a given audio recording. In the MARBLE benchmark, we utilise two datasets: Nsynth and MTG-instrument. The Nsynth dataset comprises 306,000 audio tracks, each corresponding to one of 11 different instruments. The evaluation metric for this dataset is accuracy. On the other hand, MTG-instrument is a subset of MTG-Jamendo, containing 25,000 audio tracks and 41 instrument tags. Each track can have multiple instrument tags and is evaluated based on ROC and AP.

**Singer Identification** involves recognizing the singer or vocal performer from an audio recording. In previous work on Singer Identification using the VocalSet dataset [61], different splits are employed. For the MARBLE benchmark, we randomly split the dataset into training, validation, and test sets, maintaining a ratio of 12:8:5. All sets contain the same 20 singers. The specific data divisions can be found in the provided code.

**Source Separation** aims to separate different components of a music recording, such as vocals, drums, bass, and others. In MARBLE, we adopt the widely-used MUSDB18 dataset [46] for this task. MUSDB18 consists of 150 full-length music tracks, totalling approximately 10 hours of audio and multiple isolated stems. Our training set consists of 86 tracks, the validation set contains 14 tracks, and the evaluation set comprises 50 tracks, following the official MUSDB18 setting. During training, we randomly sample 6-second segments and apply random track mixing for data augmentation. Due to the complexity of this task, we utilise the baseline architecture from the Music Demixing Challenge (MDX) 2021 [38]. This architecture consists of three linear layers and three bi-directional LSTM layers. The optimization is performed by directly computing the l2-loss between the predicted and ground-truth spectrograms. The evaluation metric for this task is the Source-to-Distortion Ratio (SDR) as defined in [38], which is calculated as the mean across the SDR scores of all songs.

## 3 Evaluation Framework

We aim to explore the generality and standardisation of the framework. Therefore, we freeze the parameters of the pre-trained model to extract pre-trained features as fixed depth embeddings fed to each downstream task-specific prediction head. This allows for as lightweight a solution as possible for all tasks, thus testing whether the representations are easily reusable across different downstream tasks. We describe pre-trained baseline models, downstream models, and protocols in the following sections.

### 3.1 Pre-trained baseline systems

The audio pre-training models explored in this paper are summarised in Table. 2. Note that we do not cover models designed entirely for speech or not open source models. We also examine all the open-source SSL systems specifically designed from music audio, in total 9 different versions of 7 pre-trained features; see Table. 2 for information on pre-trained models.

**MusiCNN** [43] is a convolutional model pre-trained on the music audio tagging task using the MSD dataset [3]. We use the default configuration of the method, which is to concatenate the mean pooling of the CNN features for a 3-second input with the output of the maximum pool.

**Contrastive learning of musical representations (CLMR)** [53] leverages a 9-layer 1-D convolutional kernel as the feature extractor, employing a number of data augmentation, and is trained on both MSD and MTT. Both are trained with a contrastive learning approach. The model extracts an embedding every 2.69 seconds.

**Jukebox** [11] is a music generation model trained using codified audio language modelling (CALM). It is trained on 1.2 million private songs, and the size of the training set is difficult to estimate the exact number of hours. However, assuming an average song length of 3-6 minutes, the total length could be 60k-120k hours, which is large and diverse to allow Jukebox to learn patterns and structures of different musical genres and styles. We use the same mid-layer representation as [8] to improve

Table 2: Information of Baseline Systems.

| Method | MusiCNN MSD-big | CLMR | Jukebox | MULE | MAP-Music2Vec | MAP-MERT-v0 base | MAP-MERT-v0 base-public | MAP-MERT-v1 base | MAP-MERT-v1 large |
|---|---|---|---|---|---|---|---|---|---|
| Network | CNN | 9-Conv | 3-Conv, 36-Trans | 22-Conv, 2-Trans | 7-Conv, 12-Trans | 7-Conv, 12-Trans | 7-Conv, 12-Trans | 7-Conv, 12-Trans | 7-Conv, 12-Trans |
| #Params | 8M | 2.5M | 5B | 62.4M | 95M | 95M | 95M | 95M | 330M |
| Input | log-mel | waveform | waveform | log-mel | waveform | waveform | waveform | waveform | waveform |
| Stride | 3s | 2.69s | 23.78s | 2s | 20ms | 20ms | 20ms | 13.3ms | 13.3ms |
| Context Length | 3s | 2.69s | 23.78s | 3s | 30s | 5s | 5s | 5s | 5s |
| Data (hour) | 10~20k | 1.7k | 60~120k | 117.5k | 1k | 1k | 0.9k | 17k | 160k |
| Pre-training Task | Music Tagging | Contrastive Learning | CALM | Contrastive Learning | MLM Boostrapping | MLM Clustering | MLM Clustering | MLM Clustering | MLM Clustering |

computational efficiency. Unlike other representations that run on short context windows, JUKEBOX is trained on a long window of 8192 sample points (23.78 seconds) of audio. We use the same strategy as [8] to extract the audio features on the downstream dataset.

**MULE (Musicnet-ULarge)** [37] is a SSL system based on **SF NFNet-F0** [58], SlowFast Normalizer-Free ResNet. It combines a SlowFast (SF) part (including a slower pathway that captures spatial information and a faster pathway that captures temporal information) with a more efficient and scalable variant of the Normalizer-Free ResNet (NFNet). MULE is contrastively pre-trained on the whole MusicSet dataset [37] and provides promising results on classification tasks. The model extracts an embedding with a 3-second window length and a 2-second hop length.

**MAP-Music2Vec** [31] is a self-supervised learning (SSL) model specifically based on a bootstrapping mask prediction pre-training strategy. It consists of two main components: the student and teacher models. Both share the same architecture with 12 transformer layers, with the teacher model's parameters being exponential moving averages of the student model's parameters. The student model takes in masked input, and during training, it aims to learn deep features from the teacher model based on the output of the unmasked input. Specifically, it computes the average of the top 8 layers of the Transformer's output in the teacher model. To train the MAP-Music2Vec model, a private dataset comprising approximately 1,000 hours of music data was used. The input length of the MAP-Music2Vec model is set to 30 seconds, producing 50 embeddings per second. These embeddings capture essential features of the music data and can be utilised for various downstream tasks, including sequential tasks such as source separation and beat tracking.

**MAP-MERT-v0**, also referred to as MERT-95M$^{\text{K-means}}$ in the work by Li et al. [29], is a pre-trained model built upon the speech self-supervised learning (SSL) system HUBERT [20]. It undergoes pre-training for masked prediction, with discrete pseudo-labels obtained from K-Means clustering on music features. The pre-training task of MAP-MERT-v0 involves two pseudo-labels based on logmel and Chroma, along with a CQT reconstruction task that emphasises pitch information. Two versions of the MAP-MERT-v0 model are included: MAP-MERT-v0[8], trained on a private dataset of 1,000 hours, and MAP-MERT-v0-public[9], trained on Music4ALL [50]. The input length of the MAP-MERT-v0 model is set to 5 seconds, generating 50 embeddings per second. This design facilitates fine-tuning for sequential tasks, enabling efficient and effective processing of music data.

**MAP-MERT-v1** encompasses two variants: (MAP-)MERT-v1-base[10] and (MAP-)MERT-v1-large[11]. These models, also known as MERT-95M$^{\text{RVQ-VAE}}$ and MERT-330M$^{\text{RVQ-VAE}}$ in the work by Li *et al*. [29], employ EnCodec, a pre-trained discrete deep feature, as a replacement for the K-means feature. This modification facilitates the scaling up of the model. Similar to MAP-MERT-v0, the input length of the MAP-MERT-v1 models is 5 seconds, but they produce 75 embeddings per second.

---

[8] https://huggingface.co/m-a-p/MERT-v0
[9] https://huggingface.co/m-a-p/MERT-v0-public
[10] https://huggingface.co/m-a-p/MERT-v1-95M
[11] https://huggingface.co/m-a-p/MERT-v1-330M

This configuration enables effective fine-tuning for sequential tasks, making the models suitable for processing music data in a variety of applications.

## 3.2 Downstreams and Training Strategies

To evaluate the relevance of representations for downstream MIR tasks, we design evaluation frameworks: the *unconstrained* track, *semi-constrained* track and the *constrained* track. In the unconstrained track, researchers are invited to submit their systems with any hyperparameter and structure configuration, including the option to fine-tune pre-trained models. This track encourages flexibility and exploration, enabling researchers to investigate a wide range of approaches. On the other hand, the semi-constrained track requires the submissions to use frozen pre-trained backbones. Finally, the constrained track employs a standardised setting with limited hyper-parameter search space (cf. Appendix A), where frozen models are used as feature extractors for training a one-layer 512-unit MLP (or 1/3-layer 512-unit LSTM for melody extraction or source separation, or 3-encoder-3-decoder layers transformer for lyrics transcription) on each task. In addition, we set a computational wall for MARBLE. The systems need to finish each task within a week on our machine equipped with a single consumer GPU (RTX3090). By offering these three evaluation tracks, we aim to provide researchers with a comprehensive platform to assess the performance and relevance of representations in MIR tasks, encouraging innovative approaches and fostering advancements in the field. For the same task with a uniform dataset, if there are different evaluation metrics (*e.g.*, emotion regression, source separation, and tagging), we will average the two evaluation metrics. We select the checkpoints regarding to the best validation results for final testing and submission.

Table 3: Performances of Baselines Evaluated on MARBLE with constrained settings (1/3). We include previous SOTAs for reference. Note that MARBLE imposes strict constraints on downstream structures and hyper-parameter search spaces, while previous SOTAs are not subject to such limitations. Best scores on MARBLE are **bold**, and best scores among all systems are underlined.

| Dataset Task | MTT Tagging | | GS Key | GTZAN Genre | GTZAN Rhythm | EMO Emotion | | Nsynth Instrument | Nsynth Pitch | VocalSet Tech | VocalSet Singer |
|---|---|---|---|---|---|---|---|---|---|---|---|
| Metrics | ROC | AP | Acc$^{Refined}$ | Acc | F1$^{beat}$ | R2$^V$ | R2$^A$ | Acc | Acc | Acc | Acc |
| MusiCNN [43] | 90.3 | 37.8 | 14.4 | 73.5 | - | 44.4 | 68.8 | 72.6 | 64.1 | 70.3 | 57.0 |
| CLMR [53] | 89.5 | 36.0 | 14.8 | 65.2 | - | 44.4 | 70.3 | 67.9 | 47.0 | 58.1 | 49.9 |
| Jukebox-5B [8, 67] | **91.4** | **40.6** | 63.8 | **77.9** | - | 57.0 | 73.0 | 70.4 | 91.6 | 76.7 | 82.6 |
| MULE [37] | 91.2 | 40.1 | 64.9 | 75.5 | - | **60.7** | 73.1 | **74.6** | 88.5 | 75.5 | __87.5__ |
| MAP-Music2Vec [31] | 90.0 | 36.2 | 50.6 | 74.1 | 68.2 | 52.1 | 71.0 | 69.3 | 93.1 | 71.1 | 81.4 |
| MAP-MERT-v0-95M [30] | 90.7 | 38.2 | 64.1 | 74.8 | __88.3__ | 52.9 | 69.9 | 70.4 | 92.3 | 73.6 | 77.0 |
| MAP-MERT-v0-95M-public [30] | 90.7 | 38.4 | **67.3** | 72.8 | 88.1 | 59.1 | 72.8 | 70.4 | 92.3 | 75.6 | 78.0 |
| MAP-MERT-v1-95M [29] | 91.0 | 39.3 | 63.5 | 74.8 | __88.3__ | 55.5 | __76.3__ | 70.7 | 92.6 | 74.2 | 83.7 |
| MAP-MERT-v1-330M [29] | 91.1 | 39.5 | 61.7 | 77.6 | 87.9 | 59.0 | 75.8 | 72.6 | __94.4__ | __76.9__ | 87.1 |
| Previous SOTA | 92.0 [21] | 41.4 [8] | 74.3 [27] | 83.5 [37] | 80.6 [19] | 61.7 | 72.1 [8] | 78.2 [58] | 89.2 [37] | 65.6 [64] | 80.3 [39] |

Table 4: Performances of Baselines Evaluated on MARBLE with constrained settings (2/3). Note that we denote the scores of *Jukebox-5B* on *MTG* tasks with asterisks(*), because it hit the computational wall of MARBLE, meaning that the system was unable to complete the corresponding task within a week on our machine equipped with a single consumer GPU (RTX3090).

| Dataset Task | MTG Instrument | | MTG MoodTheme | | MTG Genre | | MTG Top50 | | MUSDB Source Separation | | | |
|---|---|---|---|---|---|---|---|---|---|---|---|---|
| Metrics | ROC | AP | ROC | AP | ROC | AP | ROC | AP | SDR$^{vocals}$ | SDR$^{drums}$ | SDR$^{bass}$ | SDR$^{other}$ |
| MusiCNN [43] | 74.0 | 17.2 | 74.0 | 12.6 | 86.0 | 17.5 | 82.0 | 27.5 | - | - | - | - |
| CLMR [53] | 73.5 | 17.0 | 73.5 | 12.6 | 84.6 | 16.2 | 81.3 | 26.4 | - | - | - | - |
| Jukebox-5B [8, 67] | 78.5* | __22.0*__ | 77.6* | 15.3* | __88.0*__ | __20.5*__ | 83.4* | 30.4* | - | - | - | - |
| MULE [37] | 76.6 | 19.2 | **78.0** | **15.4** | **88.0** | 20.4 | **83.7** | **30.6** | - | - | - | - |
| MAP-Music2Vec [31] | 76.1 | 19.2 | 76.7 | 14.3 | 87.1 | 18.8 | 83.0 | 29.2 | 5.5 | 5.5 | **4.1** | 3.0 |
| MAP-MERT-v0-95M [30] | 76.6 | 18.7 | 75.9 | 13.7 | 86.9 | 18.5 | 82.8 | 28.8 | **5.6** | **5.6** | 4.0 | 3.0 |
| MAP-MERT-v0-95M-public [30] | 77.5 | 19.6 | 76.2 | 13.3 | 87.2 | 18.8 | 83.0 | 28.9 | 5.5 | 5.5 | 3.7 | 3.0 |
| MAP-MERT-v1-95M [29] | 77.5 | 19.4 | 76.4 | 13.4 | 87.1 | 18.8 | 83.0 | 29.0 | 5.5 | 5.5 | 3.8 | **3.1** |
| MAP-MERT-v1-330M [29] | **78.1** | 19.8 | 76.5 | 14.0 | 86.7 | 18.6 | 83.4 | 29.9 | 5.3 | **5.6** | 3.6 | 3.0 |
| Previous SOTA | 78.8 | 20.2 [1] | 78.6 | 16.1 [37] | 87.7 | 20.3 [1] | 84.3 | 32.1 [37] | 9.3 | 10.8 | 10.4 | 6.4 [48] |

## 4 Results and Discussion

According to Table 3, 4 and Fig 1, all pre-trained baseline representations on MARBLE have achieved decent results. Despite strict constraints on downstream structures and hyper-parameter search spaces,

Table 5: Performances of Baselines Evaluated on MARBLE with constrained settings (3/3). The overall average scores are calculated on the systems applicable to all tasks.

| Dataset | MelodyDB | Muljam | | Jamendo | | GuitarSet | | | | | | | |
| Task | Melody | Lyrics | | Lyrics | | Chord Estimation | | | | | | | |
| Metrics | Acc | CER | WER | CER | WER | root | majmin | mirex | thirds | triads | sevenths | majmin_inv | sevenths_inv |
| MAP-Music2Vec [31] | 36.1 | 56.4 | 87.8 | 55.7 | 89.6 | 13.7 | 11.1 | 10.4 | 10.4 | 10.4 | 11.1 | 9.4 | 9.4 |
| MAP-MERT-v0-95M [30] | 60.0 | 52.6 | 82.3 | 54.8 | 87.6 | 48.7 | 38.9 | 36.7 | 36.6 | 36.5 | 37.5 | 30.3 | **29.0** |
| MAP-MERT-v0-95M-public [30] | 36.1 | 52.5 | 82.7 | 52.6 | 85.2 | 49.1 | 38.7 | 36.4 | 36.6 | 36.4 | 37.5 | 29.9 | 28.6 |
| MAP-MERT-v1-95M [29] | 60.8 | 49.4 | 77.9 | **49.6** | **82.2** | 50.5 | 38.8 | 36.5 | 36.7 | 36.4 | 36.5 | **31.2** | 28.9 |
| MAP-MERT-v1-330M [29] | **62.5** | **48.5** | **77.0** | 50.3 | 83.1 | **56.3** | **44.8** | **45.1** | **44.0** | **44.0** | **40.7** | 27.9 | 26.5 |
| Previous SOTA | 65.3[59] | 39.5 | 54.8[44] | 25.4 | 44.4[44] | 34.8 | 33.3 | 33.6 | 33.3 | 33.2 | 24.0 | 33.1 | 23.9[22] |

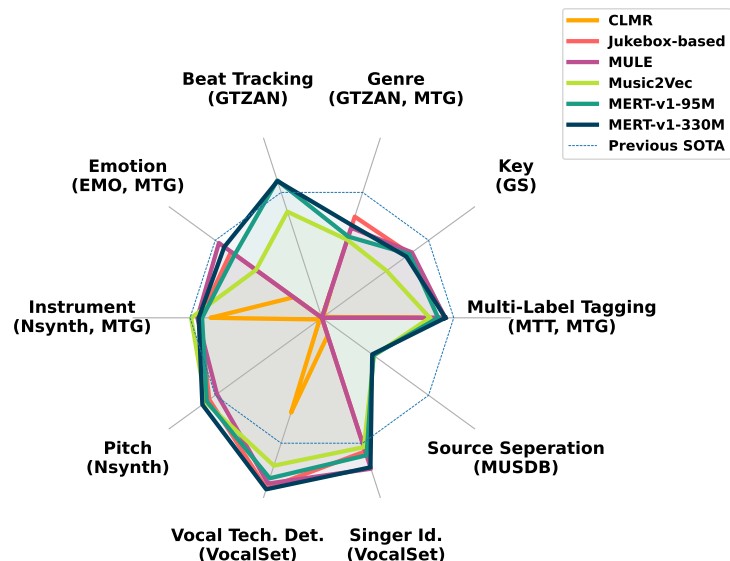

Figure 1: SSL Baselines Compared to previous SOTA. The performances of the tasks are merged according to the task types demonstrated in Tab. 1. Results not applicable are set to 0.

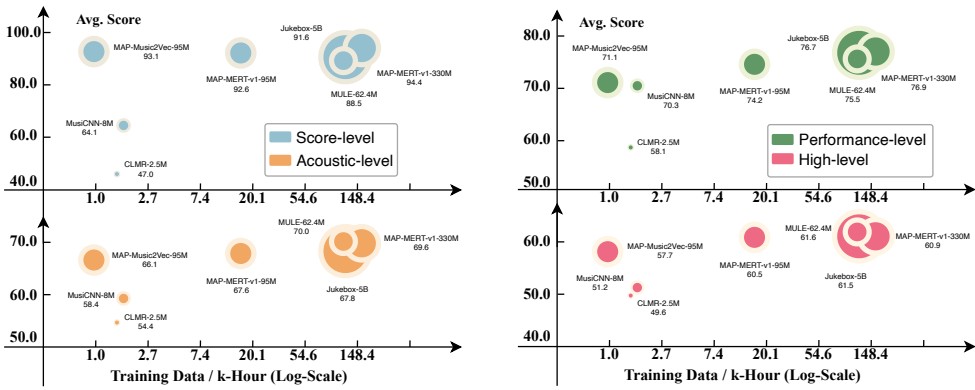

(a) Scores at Acoustic-level and Score-level.  (b) Scores at Performance-level and High-level.

Figure 2: Results Analysis Regarding to Training Data Size. Since some models are not applicable to the sequence labelling tasks, the performances of *source separation* and *beat tracking* tasks are excluded on acoustic-level and score-level average score calculation correspondingly. The radii of the scatter points are isometrically log scaling with the parameter sizes.

they are able to approach, if not surpass, the previous state-of-the-art (SOTA) in many tasks. For instance, the best performance on NSynth Pitch classification have achieved up to 94.4% accuracy. Nonetheless, the majority of tasks are still far from being solved, including music tagging and source separation tasks. Notably, the performance on MUSDB18 is merely half of the previous SOTAs.

The MAP family achieves balanced results, successfully performing tasks including sequence labelling, which other models fail to accomplish (as they do not provide frame-level representations or are too cumbersome to train). This series of models excel at multiple taxonomy levels. On certain tasks, MAP-MERTs achieve results close to or surpass the previous state-of-the-art. However, music tagging tasks are dominated by Jukebox-5B and MULE. Jukebox may benefit from its massive parameter size and generative modelling of detailed information, as well as the introduction of metadata during the pre-training period. Conversely, MULE benefits from its proprietary large-scale, high-quality dataset, MusicSet, and the highly discriminative representations learned by contrastive pre-training.

Based on Fig. 2a and 2b, excluding sequence labelling tasks (as some baselines do not support them), we observe a general trend: as the volume of data and the size of model parameters increase, the performance of tasks across four levels correspondingly improves. The choice of pre-training method and model size significantly influences the performance. For instance, MAP-Music2Vec-95M, utilizing only 1k hours of data for self-supervised learning, outperforms both supervised pre-trained MusiCNN-8M and contrastive pre-trained CLMR-2.5M on the same scale of data. More analysis could be referred to Appendix B.

## 5  Conclusion

In this work, we introduce the Music Audio Representation Benchmark for universaL Evaluation (MARBLE) as a comprehensive benchmark for evaluating pre-trained music features. It encompasses a hierarchy taxonomy that covers acoustic, performance, score, and high-level description levels, and utilises publicly available datasets for 18 MIR tasks. We establish a standardised preprocessing and data splitting protocol, along with a unified evaluation framework, to ensure fair and reproducible assessment. We report the results of all 9 open-sourced pre-trained models developed on music recordings, showcasing their performance across multiple tasks. The results demonstrate that several pre-trained models achieve comparable or even superior performance to the state-of-the-art models on various tasks within MARBLE. However, there is still ample room for improvement, particularly in music tagging and source separation. With the release of the toolkit, we hope to facilitate future research by providing easy access, reproducibility, and fair comparison of SSL pre-trained models for music understanding. We encourage engagement from researchers in the audio and AI communities to contribute to the advancement of representation learning for music information retrieval.

### Discussion and Future Work

Our benchmark has some shortcomings that can be further improved. To begin with, some of the tasks, such as beat tracking and piano transcription, typically use multiple evaluation metrics, but we only include one or two for each of the tasks due to the copyright issues preventing many datasets from being publicly available, lack of standard pre-processing or maintenance, and the limitation of time. Although the selected metric is fundamental and a good indicator, an average of all the metrics might be a better choice. Besides, some of the datasets are not sufficient for a single task. For example, the GTZAN dataset does not have a commercially-available license, and it only includes less than 10 hours of music recordings, making the evaluation more subject to bias. We will include more commercially-available larger datasets on the same tasks. Moreover, we do not include some MIR tasks that lack a common dataset currently, such as cover song detection and query-by-humming. In the future version, we will include more datasets and tasks. Last but not least, MIR on symbolic music is not included in the first version of our benchmark as well.

Apart from the traditional MIR tasks, some interesting tasks deserve more attention for benchmark development in the computer music and AIGC communities. With the benchmark and pre-trained models in MIR, developing an evaluation score on music generation and synthesis might be possible. There may not exist a perfect solution on the subject metrics for music generation to build a benchmark; otherwise, composing musical art will simply search for the waveform with the highest scores. But one can expect such a benchmark can be helpful for the music industry or music education to preclude some bad music generation. Besides, multi-modal approaches that combine music audio with symbolic music and language (*e.g.*, lyrics and music description) also deserve a benchmark.

## Acknowledgements

We would like to express sincere gratitude to our friends Anqiao Yang and Wei Fan for their invaluable support during the writing of this paper.

Ruibin Yuan is funded by Theme-based Research Scheme (T45-205/21-N) and Early Career Scheme (ECS-HKUST22201322), Research Grants Council of Hong Kong. Yinghao Ma and Jiawen Huang are research students at the UKRI Centre for Doctoral Training in Artificial Intelligence and Music, supported by UK Research and Innovation (Grant Number: EP/S022694/1). Yizhi Li is fully funded by an industrial PhD studentship (Grant Number: 171362) from the University of Sheffield, UK. Emmanouil Benetos is supported by RAEng/Leverhulme Trust Research Fellowship LTRF2223-19-106.

We acknowledge IT Services at The University of Sheffield for the provision of services for High-Performance Computing. This project also made use of time on Tier 2 HPC facility JADE2, funded by EPSRC (EP/T022205/1)

Besides, we would like to give special thanks to the following researchers or musicians on the lyrics labelling: Léo Nebel in LIP6 at Sorbonne Université; Nick Magal in School of Music at Carnegie Mellon University; Carey Bunk, Yannis Vasilakis, Christopher Mitcheltree, Nelly Victoria Alexandra Garcia-Sihuay, Teresa Pelinski Ramos, Ilaria Manco, David Südholt, Jordan Shier, and Matthew Rice in Centre for Digital Music at Queen Mary University of London; Emilian Postolache in Sapienza University of Rome; as well as Wenqin Yu in the Chinese Music Institute at Peking University.

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

# Appendix A   Evaluation Protocol Details

The hyper-parameter search range of the constrained evaluation track is given as follow:

1. **Layer**: {every single layer, weighted sum}
2. **Model**: {one-layer 512-units MLP, one-layer 512-unit LSTM (melody extraction only), 3-layer 512-unit LSTM (source separation only), 3-encoder-3-decoder layers transformer (lyrics transcription only)}
3. **Batch size**: {64}
4. **Learning rate**: {5e-5, 1e-4, 5e-4, 1e-3, 5e-3, 1e-2}
5. **Dropout probability**: {0.2}

# Appendix B   Detail Analysis

**What have the music audio pre-trained representations learned?** We observe that all the representations have learned multiple levels of knowledge in Fig. 1. Most of the selected baselines are particularly good at high-level music description tasks, such as genre classification and emotion recognition. However, when pre-trained with a full supervision paradigm, the representations may not be able to model pitch and key well, as they could overfit the supervision signal less relevant to pitch-related information. On the contrary, SSL methods usually mitigate this issue by providing more generalisable representations. Some representations do not support frame-level representations, which makes it difficult to evaluate their performance on tasks such as source-separation and beat tracking. Therefore, it is unclear how well these models have learned such information.

**How can we design better pre-training strategies for music audio representation learning?** As mentioned in the above paragraph, we suggest that a good pre-training strategy needs to prevent overfitting the supervision signal, which makes self-supervised learning a more promising approach. Moreover, we argue that an optimal method for music pre-training should be able to scale up to larger data and model size. Based on observations from Figure 2, it appears that larger data and model size have a greater impact on performance than the training paradigm (generative, contrastive, or mask prediction) at the current stage of research. Besides, stacked transformer models are good candidates for future pre-training architecture, as they can be easily scaled up, and usually provide frame-level representations in a well-considered design.

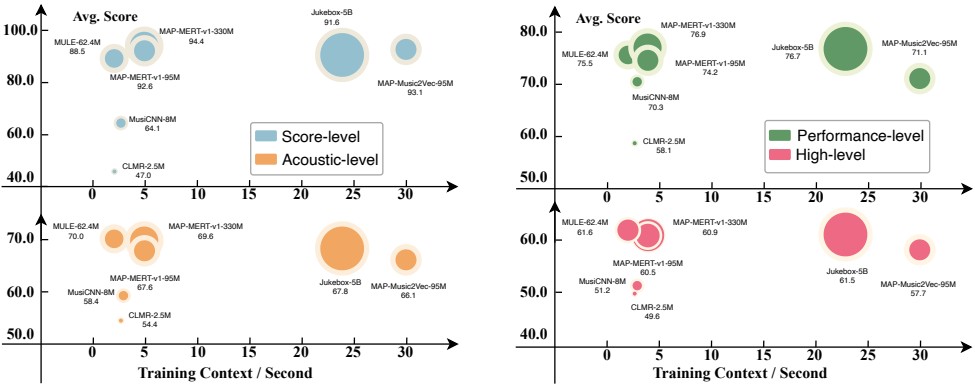

(a) Scores at Acoustic-level and Score-level.          (b) Scores at Performance-level and High-level.

Figure 3: Results Analysis Regarding Training Context Length. The performances of *source separation* and *beat tracking* tasks are ignored similar to Fig. 2.

**How does context length affect performance?** According to Fig. 3, the relationship between context length and performance exhibits a rather complex and irregular pattern, for which it is currently difficult to draw any conclusive insights. This is due to the limited number of music audio representations available at the moment, coupled with challenges in controlling variables. However, we are able to derive some preliminary observations when considering factors such as data size (D)

and parameter size (N). We observe that within a context length (L) of approximately 3 to 5 seconds, scaling up N and D can be effective, but the performance quickly saturates. Furthermore, according to MAP-Music2Vec-95M, solely increasing the L without scaling the N and D may also lead to performance saturation. Interestingly, when scaling up all three aspects, according to Jukebox-5B with 23 seconds context and 60~120khr data, the performance still saturates. The underlying cause of this saturation may be associated with the training paradigm.

## Appendix C    Website and Leaderboard

To accompany the MARBLE benchmark with leaderboard data and detailed resources presentation, we build a website, which can be found at `https://marble-bm.shef.ac.uk`. All the resources and comprehensible introduction of the benchmark and submission guideline are indexed on the homepage as shown in Fig. 4. The participants can easily find the process of submitting their results according to the guidelines. As demonstrated in Fig. 5, we provide a well-organised leaderboard for MARBLE, where the evaluated results can be re-ranked according to different metrics and filtered by tasks.

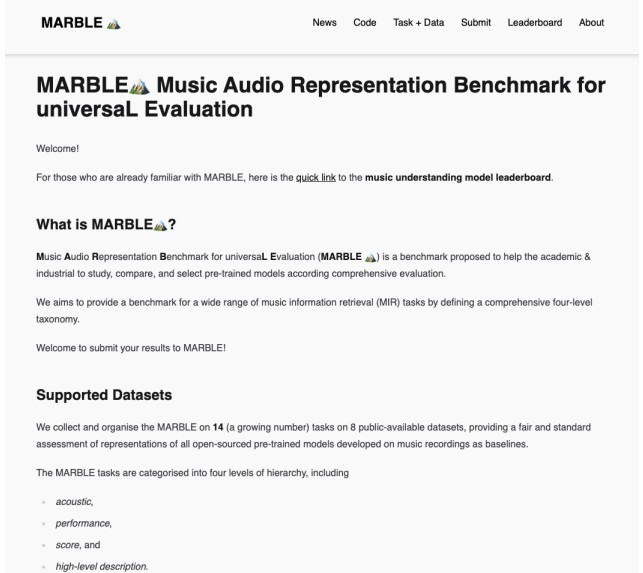

Figure 4: Website for the Proposed MARBLE Benchmark.

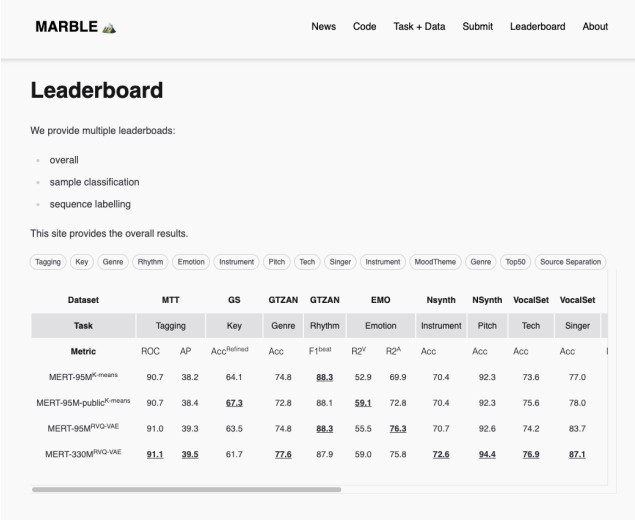

Figure 5: Music Understanding Model Leaderboard Hosted on the MARBLE Website.

## Appendix D    Details on Chord Estimation

### D.1    Chord Vocabulary

Our chord vocabulary includes "none" and 35 different chords on each of the 12 root notes, 421 in total. The root notes are listed as follows: {C Db D Eb E F Gb G Ab A Bb B}. We do not distinguish between equal notes under the twelve equal temperaments. For example, we think that C# and Db are the same note and have the essentially equivalent function in chord prediction. We use sharp in the code implementation for identification but use flat in the following tables.

The following Tables of the 35 types of chords with examples and a number of samples in the datasets.

| chord name | maj | min | aug | maj6 | min6 | 7 | maj7 | min7 | dim7 | hdim7 |
| --- | --- | --- | --- | --- | --- | --- | --- | --- | --- | --- |
| example | C:maj | C:min | C:aug | C:maj6 | C:min6 | C:7 | C:maj7 | C:min7 | C:dim7 | C:hdim7 |
| chord tones | 1,3,5 | 1,b3,5 | 1,3,#5 | 1,3,5,6 | 1,b3,5,6 | 1,3,5,b7 | 1,3,5,7 | 1,b3,5,7 | 1,b3,b5,bb7 | 1,b3,b5,b7 |
| chord number | 1120 | 368 | 16 | 70 | 12 | 374 | 292 | 204 | 2 | 106 |

| chord name | 9 | maj9 | min9 | 11 | sus2 | sus4 | maj/3 | maj/5 | min/b3 | min/5 | 7/3 | 7/5 | 7/b7 |
| --- | --- | --- | --- | --- | --- | --- | --- | --- | --- | --- | --- | --- | --- |
| example | C:9 | C:maj9 | C:min9 | C:11 | C:sus2 | C:sus4 | C:maj/3 | C:maj/5 | C:min/b3 | C:min/5 | C:7/3 | C:7/5 | C:7/b7 |
| chord tones | 1,3,5,b7,9 | 1,3,5,7,9 | 1,b3,5,b7,9 | 1,3,5,b7,9,11 | 1,2,5 | 1,4,5 | 3,5,1 | 5,1,3 | b3,5,1 | 5,1,b3 | 3,5,b7,1 | 5,b7,1,3 | b7,1,3,5 |
| chord number | 78 | 22 | 48 | 8 | 88 | 44 | 82 | 264 | 10 | 82 | 10 | 44 | 46 |

| chord name | maj7/3 | maj7/5 | maj7/7 | min7/b3 | min7/5 | min7/b7 | dim7/b3 | dim7/b5 | dim7/bb7 | hdim7/b3 | hdim7/b5 | hdim7/b7 | N |
| --- | --- | --- | --- | --- | --- | --- | --- | --- | --- | --- | --- | --- | --- |
| example | C:maj7/3 | C:maj7/5 | C:maj7/7 | C:min7/b3 | C:min7/5 | C:min7/b7 | C:dim7/b3 | C:dim7/b5 | C:dim7/bb7 | C:hdim7/b3 | C:hdim7/b5 | C:hdim7/b7 | No chord |
| chord tones | 3,5,7,1 | 5,7,1,3 | 7,1,3,5 | b3,5,b7,1 | 5,b7,1,b3 | b7,1,b3,5 | b3,b5,bb7,1 | b5,bb7,1,b3 | bb7,1,b3,b5 | b3,b5,b7,1 | b5,b7,1,b3 | b7,1,b3,b5 | No chord |
| chord number | 6 | 66 | 14 | 6 | 30 | 42 | 0 | 0 | 0 | 0 | 6 | 2 | |

These are some special or rare chords in the dataset and we use some Chord Substitutions based on similar chords or the chord annotation for the music score instead of the ground truth chord annotation the musician actually plays.

1. **majmin7** was substituted with **7**: The "majmin7" chord is equivalent to the "7" chord, so we are making a replacement to standardize the notation.

2. **minmaj7** was substituted with **min7**: Both chords share the root, minor third, and perfect fifth. When mapping minmaj7 to min7, the major seventh is altered to a minor seventh, ensuring the "minor" character of both chords remains consistent.

3. **min11** was substituted with **11**: Both chords are minor chords composed of the seventh and eleventh tones. Given their infrequent occurrences, we map "min11" to the "11" chord.

4. **Substitution for out-of-vocabulary colour chords**: The performed chord annotations in the GuitarSet also contain out-of-vocabulary colour chords such as (1,5)/1, (1,5,b7)/1, (5,2,b7,4)/4. For such chords, we identify the corresponding standard chords in the instructed chord annotations and substitute them.

5. **Special Transposition Handling for Standard Chords**: Map to the standard transposition that is closest to the corresponding transposed note.

### D.2    Chord Recognition Metric Definition

1. **root**: Evaluating chord recognition algorithms based on the root notes of the identified chords. Only compares the root of the chords.

2. **majmin**: Only compares major, minor, and "no chord" labels. Any other chord types or variations, such as 7th chords, augmented, diminished, and so on, are not considered in this specific evaluation.

3. **mirex**: Compare chords along MIREX rules. A estimated chord is considered correct if it shares at least three pitch classes in common.

4. **thirds**: Chords are compared at the level of major or minor thirds (root and third). For example, both ('A:7', 'A:maj') and ('A:min', 'A:dim') are equivalent, as the third is major and minor in quality, respectively.

5. **traids**: Chords are considered at the level of triads (major, minor, augmented, diminished, suspended). In addition to the root, the quality is only considered through #5th scale degree (for augmented chords). For example, ('A:7', 'A:maj') are equivalent, while ('A:min', 'A:dim') and ('A:aug', 'A:maj') are not.

6. **sevenths**: Compares according to MIREX "sevenths" rules. Only major, major seventh, seventh, minor, minor seventh and no chord labels are compared.

7. **majmin_inv**: Compares major/minor chords, with inversions. The bass note must exist in the triad.

8. **sevenths_inv**: Compares according to MIREX "sevenths" rules, with inversions. The bass note must exist in the chord.

During the evaluation process, frame-level predictions are directly merged to event-level by the `mir_eval` function so we do not apply any post-processing to the prediction.

## Appendix E    Details on Lyrics Transcription

### E.1    MulJam2.0 dataset

MulJam2.0 is derived from MulJam, featuring larger and more refined human annotation on the test set. We select 34 songs from the training set and obtain human lyrics annotation to expand the test set. For each language, 20 songs are randomly selected from the original training set to form the validation set. A few songs are excluded due to poor alignment for obtaining the line-level annotations (For details, please refer to [70]). We also exclude the songs in the training and validation sets that were present in Jamendo (3 songs in training and 1 song in validation), ensuring that the songs in the evaluation datasets remain unseen during training. The numbers of songs by language can be found in Tab. 6.

The human annotation is performed at the song level. We applied similar procedures to obtain line-level annotations, as was done for the training set in MulJam. We use the timestamps provided by Whisper [44], and align the lines predicted by Whisper with the human annotation. As in [70], lines with unusually high character rates (exceeding 37.5 Hz) are removed. However, for the test set we choose not to filter by the similarity between the aligned text pairs, to prevent introducing excessive bias in favor of Whisper predictions.

Table 6: Number of songs in MulJam2.0 and Jamendo datasets.

| Dataset | MulJam2.0 | | | Jamendo |
| Split | Train | Valid | Test | Test |
| --- | --- | --- | --- | --- |
| English (en) | 3557 | 20 | 28 | 20 |
| French (fr) | 977 | 19 | 19 | 20 |
| Spanish (es) | 584 | 19 | 13 | 20 |
| German (de) | 107 | 20 | 3 | 20 |
| Italian (it) | 278 | 20 | 7 | - |
| Russian (ru) | 106 | 16 | 4 | - |
| Total | 5609 | 114 | 74 | 80 |

### E.2    Language Model and Tokenizer

The language model (LM) is trained using a speechbrain [47] language model recipe [12]. The model comprises of 12 transformer encoder layers, with an attention dimension of 768, 12 attention heads, and a position-wise feed-forward layer dimension of 3072. The LM is trained using cross-entropy loss for 20 epochs, and the model with the lowest loss is selected.

The target character set is the union of the character sets from 6 languages, resulting in a total of 91 tokens: $\epsilon$, <bos>, <eos>, <unk>, A, B, C, D, E, F, G, H, I, J, K, L, M, N, O, P, Q, R, S, T, U, V, W, X, Y, Z, À, Á, Â, Ä, Æ, Ç, È, É, Ê, Ë, Ì, Í, Î, Ï, Ñ, Ò, Ó, Ô, Ö, Ù, Ú, Û, Ü, Œ, Ÿ, È, А, Б, В, Г, Д, Е, Ж, З, И, Й, К, Л, М, Н, О, П, Р, С, Т, У, Ф, Х, Ц, Ч, Ш, Щ, Ъ, Ы, Ь, Э, Ю, Я.

---

[12]https://github.com/speechbrain/speechbrain/blob/develop/recipes/LibriSpeech/LM/hparams/transformer.yaml

### E.3 Training Details

The beam search used for validation and testing incorporates a combination of CTC probabilities, LM probabilities (applied only at test time), and S2S probabilities. We assign a weight of 0.4 to the CTC probabilities and 0.3 to the LM probabilities. During validation, we utilize a beam size of 10 and calculate Word Error Rate every 5 epochs to optimize processing efficiency. For thorough evaluation, we scale up the beam size to 40 during the testing phase. The accuracy of the S2S branch output is continually monitored to determine whether early stopping should be triggered and to facilitate model selection.

### E.4 Results and Discussion

The results of multilingual lyrics transcription using different pretrained features can be found in Tab. 7. In addition to MulJam, we also present WERs on the Multilingual Jamendo evaluation set [13]. This dataset consists of 80 songs in 4 languages: English, French, Spanish, and German. While Italian and Russian songs are not included, Jamendo's human-annotated line-level annotation aligns well with our evaluation setting. For comparison, we reference the state-of-the-art model Whisper [44], a robust model designed for speech recognition but also performs effectively on singing voice. Whisper has been trained on an extensive corpus of multilingual and multitask supervised data collected from the internet. It is also the foundation of the MulJam dataset.

Lyrics transcription is a challenging task that involves detecting vocal pronunciations in the presence of background music and making the most probable predictions based on linguistic knowledge. The multilingual context makes this task even more demanding. When performing lyrics transcription with SSL features, it is essential that these features capture clear vocal information, and that the backend provides robust inference to generate coherent text from the vocal pronunciations. Achieving this with SSL features is indeed a significant challenge. The results presented in Table 7 indicate that there is room for improvement in this task.

Among the six languages we considered, English, French, and Spanish, which have a larger number of songs than the other three, yield better results. This suggests that there may be an impact from the imbalanced training data. Russian, on the other hand, produces the worst result for two main reasons: 1. Russian employs the Cyrillic writing system, which has its own set of characters. 2. The training data for Russian is insufficient for the model to establish a connection between the pronunciation rules of Cyrillic and Latin alphabets.

The MulJam test set is human-annotated at the song level but relies on the alignment with Whisper results to derive line-level annotations. Therefore, it is worth noting that bias is introduced, as the alignment is reliable only when the human annotation closely matches the Whisper's prediction.

Table 7: Multilingual lyrics transcription results on MulJam and Jamendo.

| Language | English | | French | | Spanish | | German | | Italian | | Russian | | Whole | |
| Metric | CER | WER | CER | WER | CER | WER | CER | WER | CER | WER | CER | WER | CER | WER |
|---|---|---|---|---|---|---|---|---|---|---|---|---|---|---|
| | | | | | | MulJam2.0 test | | | | | | | | |
| MAP-Music2Vec [31] | 54.7 | 79.2 | 58.3 | 90.9 | 43.2 | 83.7 | 63.4 | 99.5 | 53.0 | 91.9 | 101.6 | 125.6 | 56.4 | 87.8 |
| MAP-MERT-v0-95M [30] | 48.7 | 71.2 | 55.5 | 85.4 | 41.0 | 80.1 | 65.9 | 100.9 | 49.1 | 86.3 | 99.5 | 124.9 | 52.6 | 82.3 |
| MAP-MERT-v0-95M-public [30] | 49.0 | 71.2 | 55.3 | 85.4 | 39.0 | 76.6 | 63.5 | 99.9 | 50.3 | 90.3 | 104.7 | 129.3 | 52.5 | 82.7 |
| MAP-MERT-v1-95M [29] | **45.5** | 66.5 | 52.5 | 81.9 | 38.2 | 73.9 | 58.8 | 93.2 | 44.4 | 81.6 | **96.1** | **117.8** | 49.4 | 77.9 |
| MAP-MERT-v1-330M [29] | **45.5** | 65.9 | 50.7 | 79.6 | 35.9 | 71.9 | 58.3 | 93.1 | 42.4 | 80.3 | 100.5 | 125.5 | 48.5 | 77.0 |
| SOTA [44] | 33.2 | 44.8 | 52.9 | 70.1 | 29.9 | 43.8 | 36.5 | 53.0 | 38.1 | 58.5 | 34.7 | 53.7 | 39.5 | 54.8 |
| | | | | | | Jamendo | | | | | | | | |
| MAP-Music2Vec [31] | 49.0 | 73.6 | 55.3 | 87.1 | 50.3 | 90.7 | 67.8 | 108.8 | - | - | - | - | 55.7 | 89.6 |
| MAP-MERT-v0-95M [30] | 48.5 | 71.8 | 54.0 | 85.1 | 49.3 | 87.6 | 67.6 | 108.1 | - | - | - | - | 54.8 | 87.6 |
| MAP-MERT-v0-95M-public [30] | 46.9 | 71.5 | 52.0 | 81.5 | 44.8 | 82.8 | 66.3 | 106.8 | - | - | - | - | 52.6 | 85.2 |
| MAP-MERT-v1-95M [29] | **43.6** | **67.2** | **49.4** | **79.6** | 43.2 | 80.6 | 62.1 | 103.3 | - | - | - | - | **49.6** | **82.2** |
| MAP-MERT-v1-330M [29] | 45.7 | 68.8 | 50.2 | 80.1 | 44.1 | 82.8 | **61.0** | **102.3** | - | - | - | - | 50.3 | 83.1 |
| SOTA [44] | 24.9 | 39.3 | 29.2 | 49.9 | 21.2 | 41.7 | 25.8 | 46.6 | - | - | - | - | 25.4 | 44.4 |

