# OpenReview forum: "MARBLE: Music Audio Representation Benchmark for Universal Evaluation"
_NeurIPS.cc/2023/Track/Datasets_and_Benchmarks — NeurIPS 2023 Datasets and Benchmarks Poster_

### Official Review · Reviewer_Gtoe · 2023-07-18
**The music audio representation benchmark for coarse-grained discriminative music processing tasks.**

**Rating:** 5
**Confidence:** 4
**Correctness:** Yes.
**Clarity:** Yes.

**Strengths:**

1. learning msuci audio representation for evalutaion of discriminative music tasks is important and interesting.

**Additional Feedback:**

In the weakness.

**Documentation:**

Yes.

**Ethics:**

None.

**Limitations:**

None.

**Opportunities For Improvement:**

1. The authors only include the coarse-grained discriminative music processing task, and it is better for the authors to provide the fine-grained music processing tasks.
2. The title of this paper is music audio representation, and it is better for the authors to include both discrimiantive and generative tasks as the benchmark.
3. For some tasks, the size of the datasets is very limited, and it is better for the author to construct the larger one for the benchmark.


**Relation To Prior Work:**

The previous work focus on the single tasks, and this paper unifed 14 tasks in the benchmark, which is useful for the music processing research.

**Summary And Contributions:**

The authors introduce the music audio representation benchmark for universal evaluation. The authors establish the unified protocol based on 14 tasks on 8 public-available datasets, defined with four levels of hierarchy, including acoustic, performance, score and high-level description. In this paper, their benchmark is only basd on discriminative tasks, while the music processing involces both discriminative and generative tasks. It is better for the authors to provide both discrimiantive tasks and generative tasks for music audio representation. For the description tasks, the authors only provide four coarse-grained tasks, including key detection, music tagging, genre classification and emotion recognition. It is better for the authors to provide the fine-grained descrition tasks such as music transcription, which is more important for music processing. For the score-level tasks, it is also better for the authors to provide the fine-grained pitch classification task based on phoneme-level, which is more important for music processing. Overall, it is better for the authors to provide the fine-grained discriminative tasks for music processsing and include the generative tasks.

---

> ### Author Response · Authors · 2023-08-23
> **Rebuttal**
>
> We appreciate the thoughtful feedback on our work.
>
> **Fine-Grained Music Processing Tasks:**
> We appreciate the suggestion to include fine-grained music-processing tasks. Your reference seems to lean towards frame-level prediction or regression as opposed to clip-level prediction. We'd like to highlight that our benchmark already encompasses fine-grained tasks: beat tracking and source separation, both in terms of classification and generation. Upcoming frame-level classification tasks will encompass performance recognition on Chinese flute and Kyoto, multi-lingual lyrics transcription, and singing pitch transcription. One of the preliminary result are as follows:
>
> |                | Guzheng f1-score | Dizi f1-score |
> |----------------|------------------|---------------|
> | sota           | 69.3             | 79.9          |
> | probe MERTv1-95M| 60.3            | 70.3          |
> | finetune MERTv1-95M| 81.8         | 82.8          |
>
>
> **Why not music generation:**
> Please refer to our **Reply on common concerns** at the top.
>
>
> **Why not new dataset:**
> Please refer to our **Reply on common concerns** at the top.
>
>
> **Dataset Size:**
> - The challenge with labeled music datasets is their inherent size constraint due to music copyright concerns and annotation costs. Acquiring music audio, especially with redistribution rights, can be exorbitantly costly. Moreover, numerous tasks demand expensive annotations from professional musicians. To this end, one of the goals for MARBLE is to test and promote those pretrained representations that require less downstream data size and able to generalize.
> - In our benchmark, tasks like music tagging, genre classification, emotion detection, pitch classification, and instrument classification have relatively more affordable annotation costs, with the dataset spanning hundreds of hours of music recordings. The key detection, for instance, contains 56 hours of recordings, deeming it sufficiently large.
> - For tasks like source separation, singer identification, etc., legally available datasets are scarce. Using the example of source separation, the MUSDB18 dataset is the sole accessible one for us. Given these constraints, we've done our best to incorporate the most relevant datasets and are continually exploring more.

---

### Official Review · Reviewer_Rutv · 2023-07-21
**A comprehensive benchmark on music understanding but lacks good insights**

**Rating:** 6
**Confidence:** 2
**Correctness:** The claims are correct
**Clarity:** The paper is readable.

**Strengths:**

The paper provides a strong foundation for comparing models with comprehensive task structures and unified protocols. The authors not only provide the toolkit but also establish a leaderboard for the community, promoting transparency, reproducibility, and further advancement in the field.

**Additional Feedback:**

N/A

**Documentation:**

Documentation is sufficient.

**Ethics:**

No noticeable ethical concerns.

**Limitations:**

1. The benchmark's ability to generalize across all types of music (genres, periods, cultures) isn't clear from the summary. Some music might be underrepresented or not represented at all in the existing datasets. Perhaps a table showcasing the diversity of each dataset could help.
2. The benchmark lacks solid analysis and only presents performance comparisons. Some key analyses for example effect of noise, data augmentations, etc could be helpful.
3. While the benchmark could be useful for academic research, its effectiveness and relevance for real-world applications in the industry are not discussed.

**Opportunities For Improvement:**

All the datasets used are from clean data. An analysis of noise robustness would help the benchmark stand out. Specifically, analysis with background noise and label noise typical in crowdsourced data may be more insightful for deployment scenarios.

**Relation To Prior Work:**

Prior work is discussed.

**Summary And Contributions:**

The paper introduces MARBLE, a benchmark for Music Information Retrieval (MIR) tasks that aim to advance AI's understanding of music. It provides a comprehensive taxonomy across four levels: acoustic, performance, score, and high-level description, and assesses open-source pre-trained music models on 14 tasks using 8 publicly available datasets. Results indicate that pre-trained musical language models perform best but have room for improvement. The authors have published a leaderboard and toolkit repository to foster further music AI research.

---

> ### Author Response · Authors · 2023-08-23
> **Rebuttal**
>
> Thank you for your valuable feedback. We appreciate the detailed observations and recommendations. We'd like to provide further clarity and address the raised concerns sequentially:
>
> 1. **Dataset Diversity**:
>
>     Our benchmark currently encompasses both vocal and instrumental music, spanning real-time recordings and synthesized datasets. They cover a range of genres and periods: from 19th-century classical pieces to modern experimental tracks. Prominently, the MTG-Jamendo dataset, a comprehensive and diverse source, plays a pivotal role in our evaluation. For more specifics, you may refer to [MTG-Jamendo](https://mtg.github.io/mtg-jamendo-dataset/).
>
>     We acknowledge our dataset's western-centric focus and are actively working to enhance diversity. Presently, we are integrating datasets that capture the nuances of Chinese musical instruments like the Dizi and Guzheng. Preliminary results, as shown below, suggest that while models pre-trained on Western music may not immediately excel on these diverse datasets, their learned parameters offer a solid foundation for fine-tuning.
>
>     |         | Guzheng f1-score | Dizi f1-score |
>     |---------|------------------|---------------|
>     | sota    | 69.3             | 79.9          |
>     | probe MERTv1-95M | 60.3 | 70.3 |
>     | finetuneMERTv1-95M | 81.8 | 82.8 |
>
> 2. **Depth of Analysis**:
>
>     We've conducted in-depth analysis of our results, detailed in Section 4, highlighting the specific strengths of various model types. For instance, the MAP family demonstrates balanced performance across tasks, while Jukebox-5B excels in music tagging. Additionally, we explore the influence of data volume, model size, context length, and pre-training strategy. These insights aim to guide the design of improved pre-trained LMs for music.
>
>     Regarding noise analysis, we believe that it can give useful insights for MARBLE, but as the first general music understanding benchmark, our current focus is adding more tasks to MARBLE. We may delve into noise's effect in our future updates.
>
> 3. **Data Augmentation**:
>
>     Our main objective was to evaluate the robustness of self-supervised features without the influence of data augmentation. Thus, in the constrained setting, we abstained from augmentations. However, in the unconstrained setting, researchers are free to employ custom enhancements. We'll accentuate this differentiation more explicitly in our work.
>
> 4. **Industry Relevance**:
>
>     The selected tasks are industry-relevant MIR tasks such as music tagging, beat tracking, and source separation. These have direct applications in music recommendation systems, karaoke, tiktok video jamming, and other end-user services.
>
>     Although many pre-trained features by major corporations are proprietary (e.g. https://ieeexplore.ieee.org/stamp/stamp.jsp?tp=&arnumber=9414405), our benchmark aims to offer a neutral ground for performance comparisons. Such benchmarks have proven invaluable in fields like NLP (GLUE) and speech (SUPERB), and we aspire to provide a similar platform for the music audio domain.

---

> > ### Comment · Reviewer_Rutv · 2023-08-26
> > **Response to Authors**
> >
> > I would like to thank the authors for their detailed responses. Regarding data augmentation and industry relevance, I believe my concerns were addressed. However, I still believe diversity is a major weak point in this dataset but the inclusion of Dizi and Guzheng is a step in the right direction. The authors should also consider showcasing diversity across music of different eras and genres. Furthermore, while adding more tasks certainly adds value to the benchmark, my opinion is that more in-depth analysis of noise and augmentation techniques will provide better insight for researchers. However, I understand the authors' desired focus on adding more tasks and leaving further analysis as future work. In that case, I would suggest adding a few lines in the paper suggesting best practices for deployment for each task based on the analyses done.

---

> > > ### Author Response · Authors · 2023-08-28
> > > **Reply by authors**
> > >
> > > **About Dataset Diversity:**
> > >
> > > Thank you for your constructive feedback on the diversity aspect of our dataset. We agree that a table summarizing the diversity of the music genres, periods, and cultures in our benchmark would be valuable. We will add it to our appendix in the next revision. However, as the first general benchmark for music understanding, our immediate focus remains on expanding the diversity of tasks, primarily those based on mainstream music. Our long-term objective is, of course, to also include a broader range of music from various cultures, eras, and genres to make the benchmark truly comprehensive.
> > >
> > > **About Best Practice Guidelines:**
> > >
> > > We appreciate your suggestion on adding guidelines for best practices for each task. We understand that these could provide valuable insights for researchers looking to apply these benchmarks in practical settings. We plan to include the best practices guidelines in the appendix of our revised paper.

---

> > > > ### Comment · Reviewer_Rutv · 2023-08-28
> > > >
> > > > Thanks for the response. I have no further concerns and I have updated my score

---

### Official Review · Reviewer_NaXM · 2023-07-21
**Review for MARBLE: Music Audio Representation Benchmark for Universal Evaluation**

**Rating:** 5
**Confidence:** 3
**Correctness:** The paper seems correct and claims we…
**Clarity:** The paper is well-written and easy to…

**Strengths:**

- Incorporating a wide range of MIR tasks and datasets and providing benchmark (MARBLE) to facilitate comprehensive music model evaluation.
- Designing a unified assessment protocol and building corresponding evaluation suites for processing, training, and benchmarking.
- These two components could make a valuable contribution to the MIR research community.

**Additional Feedback:**

L319, It would be beneficial to reference the paper (https://arxiv.org/abs/2211.09385) published at NeurIPS 22' that introduced subjective metrics for music generation.

**Documentation:**

Yes.

**Ethics:**

No noted ethical concerns.

**Limitations:**

In my opinion, the suggested benchmark would be beneficial to the MIR research community. However, it only collects existing tasks and datasets, which limits its originality. It would be better to introduce a new baseline for the proposed benchmark or provide an analysis of that baseline across various tasks and metrics in MARBEL.
Section 4 discusses benchmarks. If the analysis is expanded upon and a baseline based on the analysis that excels in the entire benchmark is presented, the paper will hold more significance.


**Opportunities For Improvement:**

- Incorporating a wide range of MIR tasks and datasets and providing benchmark (MARBLE) to facilitate comprehensive music model evaluation. -> only deal with discriminative tasks; it could be better to extend for generative tasks such as music generation.
- Designing a unified assessment protocol and building corresponding evaluation suites for processing, training, and benchmarking.
- These two components could make a valuable contribution to the MIR research community.
-> In my opinion, the suggested benchmark would be beneficial to the MIR research community. However, it only collects existing tasks and datasets, which limits its originality. It would be better to introduce a new baseline for the proposed benchmark or provide an analysis of that baseline across various tasks and metrics in MARBEL.

**Relation To Prior Work:**

The relation to prior work and contribution are discussed.

**Summary And Contributions:**

This paper introduces a comprehensive benchmark, Music Audio Representation Benchmark for universaL Evaluation (MARBLE), for evaluating Music Information Retrieval (MIR) tasks. Also, we design a unified protocol and build tool-kits in MARBLE to evaluate the generalization ability of the models. It unified the assessment protocol and built corresponding evaluation suites for processing, training, and benchmarking.

---

> ### Author Response · Authors · 2023-08-23
> **Rebuttal**
>
> Thank you for your valuable feedback. We appreciate the detailed observations and recommendations. We'd like to provide further clarity and address the raised concerns sequentially:
>
> **Why not music generation:**
> Please refer to our Reply on common concerns at the top.
>
> **Why not new dataset:**
> Please refer to our Reply on common concerns at the top.
>
> **No new baselines?**
> 1. MARBLE stands out as a novel music understanding benchmark. While it collects existing tasks, it's noteworthy that no previous models have been evaluated in such a comprehensive and unified setting. Most models have only been evaluated against a subset of these tasks.
>
> 2. Our benchmark does not only incorporate established models but also includes recent developments, such as MERT and music2vec. They can be considered as new.
>
> 3. Lastly, the cost of pretraining a universally applicable SSL representation is substantial. For the scope of a benchmark, this would not be appropriate or feasible.

---

> > ### Comment · Reviewer_NaXM · 2023-08-30
> >
> > Dear authors,
> >
> > I appreciate your responses.
> > But I still have a concern about the originality of this work.
> > Thank you so much.
> >
> > Best wishes, Reviewer NaXM.

---

### Official Review · Reviewer_XGB3 · 2023-07-22
**Benchmarking framework for music representation learning**

**Rating:** 6
**Confidence:** 5
**Correctness:** definitely correct
**Clarity:** very well written and easy to follow

**Strengths:**

- the first paper proposing a benchmarking framework for music representation learning
- covers a lot of discriminative tasks in music (MIR tasks) as well as dense prediction tasks like source separation (not covered in any prior work)
- metrics proposed follow the state-of-the-art setting, e.g., global SDR for music separation
- evaluation done on a large-scale model like Jukebox with CALM setting
- well written and easy to follow

**Additional Feedback:**

please refer to the above comments

**Documentation:**

Very likely that the paper gives enough detail

**Ethics:**

I did not find any ethical concern

**Limitations:**

- not many music tasks other than MIR tasks (only source separation in the paper) is introduced in the paper, which somehow limits the interest of readers
- only the best used metric for each task is included, e.g., global SDR for source separation, which can only highlight one aspect of the model performance

**Opportunities For Improvement:**

- a lot of attentions for music generation in ML community, which was not covered in the paper without clear explanation
- Jukebox is trained with the signal reconstruction criteria using autoencoders and good at generation, therefore not always fair to compare it with models trained with contrastive learning that should be able to capture more semantically meaningful embeddings. The authors should state the difference in how the models are trained should be clarified in the paper

**Relation To Prior Work:**

there are some prior works that addressed benchmarking for music representation learning, e.g., CALM paper, but no paper has ever proposed a benchmarking framework for music representation learning, which includes dense prediction tasks like source separation

**Summary And Contributions:**

This paper proposes a benchmarking framework for music representation learning. The focus of the proposed benchmark is on discriminative tasks while some dense prediction tasks like source separation are included. Evaluations using the framework were conducted over a number of methods including a large-scale model Jukebox with CALM setting.

---

> ### Author Response · Authors · 2023-08-24
> **Rebuttal**
>
> Thank you for your insightful comments and critiques. We appreciate the opportunity to clarify and address the issues you've raised.
>
> **Why not music generation:**
> Please refer to our **Reply on common concerns** at the top.
>
> **Fairness of Including Jukebox for Comparison:**
>
> 1. Although Jukebox is trained using signal reconstruction criteria via autoencoders, its embeddings have demonstrated robust performance in understanding tasks. For instance, the CALM paper and our benchmark tests show that Jukebox excels in some of the understanding tasks.
>
> 2. Regarding fairness, it's worth noting that Jukebox has a significantly larger number of parameters compared to other baselines. This allows Jukebox to learn a richer set of features.
>
> **Model Descriptions:**
> We already have included detailed model descriptions in Table 2.
>
> **Limited Non-MIR Tasks:**
> While we recognize that extending the task list beyond MIR tasks may widen the paper's appeal, our primary focus is on providing a comprehensive benchmark for MIR tasks, including source separation, as elaborated below.
>
> **Inclusion of Source Separation as an MIR Task:**
> We consider source separation to be part and parcel of MIR tasks. MIR encompasses a wide range of tasks related to the retrieval and understanding of music information, including source separation. As such, its inclusion in MARBLE is in line with our primary objective of providing a benchmark for MIR tasks.
>
> **Selection of Metrics:**
> We acknowledge your point about the importance of diversified metrics for comprehensive evaluation. In the initial version of MARBLE, our aim was to cover four taxonomies with a wide range of tasks using consolidated metrics cited in existing literature. Due to resource constraints, we focused on implementing the most commonly reported metrics for each task to facilitate direct comparison of baseline performances. We will consider incorporating a broader array of metrics in future versions and reporting them on our leaderboard.

---

> > ### Comment · Reviewer_XGB3 · 2023-08-29
> >
> > Thank you for preparing rebuttal comments.
> >
> > Unfortunately, I could not find any solution relevant to my proposals to improve the paper. So I leave my score as it is.

---

### Official Review · Reviewer_fqYR · 2023-07-31
**A new music audio representation benchmark**

**Rating:** 5
**Confidence:** 3
**Correctness:** Yes
**Clarity:** Yes

**Strengths:**

+ MARBLE provides a comprehensive benchmark for Music Information Retrieval (MIR) tasks. It defines a four-level taxonomy and establishes a unified protocol across 14 tasks and 8 public datasets.

+ It serves as an evaluation tool and can be easily used to measure the performance of different music audio pre-training approaches.

**Additional Feedback:**

See Opportunities For Improvement.

**Documentation:**

No. The source codes of the work have not been released well.

**Limitations:**

The authors discussed the method limitations in the paper.

**Opportunities For Improvement:**

+ The authors have not released the code library for the experiments shown in the work. This is a major limitation, as it prevents other researchers from replicating the results and building upon the work. I believe that the code library is the most valuable part of this work, and without it, I cannot recommend acceptance.

+ The scientific contribution of this work is limited. The authors did not collect any new datasets, and the approaches they use are all existing.

+ I understand that the paper covers a lot of tasks and methods, and it is not possible to provide full details in the main text. However, I do not understand why the authors did not provide these details in the appendix. The appendix is a perfect place to provide additional information, such as the code library, the datasets, and the experimental setup.

**Relation To Prior Work:**

Yes

**Summary And Contributions:**

The paper introduces the Music Audio Representation Benchmark for universaL Evaluation (MARBLE), addressing the gap in music understanding within AI. MARBLE provides a comprehensive benchmark for Music Information Retrieval (MIR) tasks, defining a four-level taxonomy and establishing a unified protocol across 14 tasks and 8 public datasets. The platform serves as a standardized evaluation tool for pre-trained music models and offers a clear, extendable framework for the community.

***Post-Rebuttal***

Thanks for the reply from the authors. The rebuttal cannot fully address my major concerns. Since the authors did not collect and annotate any new data or propose any new approaches, the contribution of this work is relatively marginal.

I keep my rating unchanged.

---

> ### Author Response · Authors · 2023-08-22
> **Rebuttal**
>
> We appreciate the thoughtful feedback on our work.
>
> **1. Code and replicability:**
> - Please refer to our **Reply on common concerns** at the top.
>
> **2. Why not new datasets:**
> - Please refer to our **Reply on common concerns** at the top.
>
> **3. Detailed Documentation:**
> - We believe we have summarized the eval protocol in the appendix, which follows a unified design. Details about tasks and datasets are also provided in Sec. 2 of the paper.
> - We will soon update a detailed README in our codebase, to describe related information in detail, including current task designs, how to add a new task, etc. MARBLE will offer sustained community support and constant updates with new tasks and datasets.

---

### Author Response · Authors · 2023-08-22
**Reply on common concerns**

Thank you to the reviewers for their detailed feedback and constructive comments on our work. We would like to address common concerns, ensuring clarity and providing necessary context where needed.

**1. Code and replicability (for reviewer fqYR):**
- It seems there might have been an oversight. Our codebase has always been accessible through the benchmark's official website. The navigation bar at the top of the site has a link to the code, which directs users to our GitHub codebase ([link](https://github.com/a43992899/MARBLE-Benchmark)). The URL for the official website has been provided both in the paper and the openreview system.
- Our codebase offers dataset downloads, preprocessing scripts, and automatic model downloads. Each task comes with a yaml template that meticulously records a variety of hyperparameters to ensure reproducibility, making it user-friendly.
- We will highlight the codebase pointer in more visible locations within the paper and website to avoid further confusion.

**2. Why not new datasets (for reviewer fqYR, NaXM):**
- We appreciate the reviewers bringing up the concern regarding new dataset creation. We too wanted to contribute additional datasets to the domain of audio music understanding. However, after extensive internal discussions, we decided to prioritize representative existing datasets that do not pose copyright risks.
- The music industry's stance on copyright has been stringent and historically well-established. For a benchmark like MARBLE, which we hope to update long-term and offer sustained community support, we invested significant effort into understanding dataset copyrights. We endeavored to include datasets with public access to audio that present minimal copyright concerns, ensuring diversity in datasets and tasks.
- Purchasing substantial music copyrights for an academic benchmark is not feasible, and ignoring copyright risks entirely is not appropriate for MARBLE. Opting to follow popular datasets (e.g. MillionSongDataset, SecondHandSongDataset) that only release metadata without providing audio downloads was not our choice due to the inherent risks with non-royalty-free music data and potential broken links (i.e. YouTube video links in the metadata) over time, leading to reproducibility issues.
- Furthermore, the primary focus of this work is on benchmarking rather than dataset creation. Notable benchmarks in other domains, like GLUE in NLP, SUPERB in speech, and HEAR in audio, also did not introduce their datasets. For a benchmark, our emphasis was predominantly on evaluation.

**3. Why not music generation (for reviewer XGB3, NaXM, Gtoe):**
- We recognize the significant interest in music generation within the community. However, MARBLE is fundamentally geared towards music representation learning evaluation, focusing primarily on understanding tasks.
- Given that the evaluation for music generation remains an open question, we believe that an effective benchmark for general music understanding can concurrently propel research into music generation evaluation. Generation needs understanding, and the community doesn’t even have a general music understanding benchmark.
- We will ensure clearer communication both in the paper and the website about MARBLE's primary focus on representation learning and music audio understanding.

---

### Comment · Area_Chair_d8rF · 2023-08-29
**Discussion**

Dear reviewers,

The authors have uploaded their rebuttal. Please take time to go over it. If you have any further questions or concerns regarding the authors' rebuttal, please start a discussion. If you are willing to adjust your scores after reading the rebuttal, please do. For those who have already done it, thanks!

Best,

AC

---

### Decision · Program_Chairs · 2023-09-22

**Decision:**

Accept (Poster)

**Comment:**

I believe that this work is of interest and sufficient quality for the community to be presented in this track. I don't see any strong reason for rejecting it, especially among the reviews.